# Self-Assessment Tests are Unreliable Measures of LLM Personality

**Akshat Gupta[1], Xiaoyang Song[2], Gopala Anumanchipalli[1]**
[1]UC Berkeley, [2]University of Michigan
akshat.gupta@berkeley.edu

## Abstract

As large language models (LLM) evolve in their capabilities, various recent studies have tried to quantify their behavior using psychological tools created to study human behavior. One such example is the measurement of "personality" of LLMs using self-assessment personality tests developed to measure human personality. Yet almost none of these works verify the applicability of these tests on LLMs. In this paper, we analyze the reliability of LLM personality scores obtained from self-assessment personality tests using two simple experiments. We first introduce the property of *prompt sensitivity*, where three semantically equivalent prompts representing three intuitive ways of administering self-assessment tests on LLMs are used to measure the personality of the same LLM. We find that all three prompts lead to very different personality scores, a difference that is statistically significant for all traits in a large majority of scenarios. We then introduce the property of *option-order symmetry* for personality measurement of LLMs. Since most of the self-assessment tests exist in the form of multiple choice question (MCQ) questions, we argue that the scores should also be robust to not just the prompt template but also the order in which the options are presented. This test unsurprisingly reveals that the self-assessment test scores are not robust to the order of the options. These simple tests, done on ChatGPT and three Llama2 models of different sizes, show that self-assessment personality tests created for humans are unreliable measures of personality in LLMs.

## 1 Introduction

As large language models (LLM) evolve and scale up (Radford et al., 2018, 2019; Brown et al., 2020; Ouyang et al., 2022; OpenAI, 2022, 2023; Zhang et al., 2022; Touvron et al., 2023a,b), they are now being used to augment humans in many different domains. For example, LLMs are being used as creative writers (Yuan et al., 2022), as educators (Jeon and Lee, 2023), as personalized assistants (Chen et al., 2023) and in many other scenarios (Eloundou et al., 2023). As more use cases of LLMs emerge every day, it has now become important to analyze and measure the behavior of such models. While LLMs now go through safety training to prevent harmful behavior (OpenAI, 2022, 2023; Touvron et al., 2023b), the measurement of behavior of such models is still not an exact science.

Personality in humans as defined by the American Psychological Association is an enduring characteristic and behavior that comprise a person's unique adjustment to life (Association, 2023). Numerous recent studies have naively tried to measure personality in LLMs using self-assessment tests created to measure human personality (Karra et al., 2022; Jiang et al., 2022; Miotto et al., 2022; Song et al., 2023; Caron and Srivastava, 2022; Huang et al., 2023; Bodroza et al., 2023; Safdari et al., 2023; Pan and Zeng, 2023; Noever and Hyams, 2023). Self-assessment tests for humans contain a list of questions where a test taker usually responds to a situation by rating themselves on a Likert-type scale (Likert, 1932), typically ranging from 1 to 5 or 1 to 7. Examples of such questions are given in Table 2. While these self-assessment tests have been shown to be reliable measures of personality for humans (Digman, 1990; Goldberg, 1990, 1993), the direct applicability of these tests for measuring LLM personality cannot be taken for granted.

Answering self-assessment questions is a non-trivial task and requires a heterogeneous combination of different steps, including understanding the question, finding the correct answer, and then projecting the answer on the given scale. As LLMs are put through these self-assessment tests, many things can go wrong in each of these steps. Thus, to even consider using these tests to measure LLM behavior, we must first evaluate the applicability of these self-assessment tests for the personality measurement of LLMs. To the best of our knowledge,

only one prior work (Safdari et al., 2023) tries to verify this. By calculating different metrics, Safdari et al. (2023) conclude the personality scores calculated using self-assessment tests are valid and reliable. We argue against those conclusions using two straightforward experiments. Our argument is based on the fact that LLMs are different from humans, thus any test that checks the validity of these self-assessment tests on LLMs must also evaluate characteristics unique to LLMs.

In this paper, we perform two insightful experiments to check the reliability of self-assessment test results for the personality measurement of LLMs. In the first experiment, we evaluate the model's ability to understand different forms of asking the same assessment question (**Prompt Sensitivity**). The hypothesis here is that input prompts that are semantically similar should lead to similar test results. In this step, we do not try to engineer prompts to trick the model. Instead, we adopt the exact same prompt template used in three previous studies (Jiang et al., 2022; Miotto et al., 2022; Huang et al., 2023) to ask assessment questions (Table 1). We find that the three semantically equivalent prompts used to ask the same personality test question lead to very different personality scores for the same model, and these differences are statistically significant. This casts doubt on the reliability of the obtained personality scores in previous works and their conclusion that personality exists in LLMs (Jiang et al., 2022) as none of them use multiple equivalent prompts to evaluate the personality of the same model.

In the second experiment, we test the sensitivity of test responses to the order in which the options are presented to the model (**Option-Order Sensitivity**). Previous studies (Robinson et al., 2022; Pezeshkpour and Hruschka, 2023) have shown that LLMs are sensitive to the order in which the options are presented in multiple-choice questions (MCQ) and are more likely to select certain options over others, irrespective of the correct answer. As self-assessment tests usually exist in the form of multiple choice questions (MCQ), we check the sensitivity of the test scores to the order of options. Specifically, we invert the order of the options or the direction of scale provided to answer the test questions. We find that the test scores have differences that are statistically significant for different presentations of option orders or direction of scale. This is in contrast to studies in humans (Rammstedt and Krebs, 2007; Robie et al., 2022) which show that human personality test results are invariant to the order in which the options are presented.

We perform these experiments on chat models as these models are aligned to produce responses in a conversational format. We specifically do these experiments with ChatGPT (OpenAI, 2022) and three Llama2 (Touvron et al., 2023b) models. We want the readers to note that although the three Llama models belong to the same model family, they are very different behaviorally as can be seen in this paper. Since LLMs are not humans and have their own unique characteristics like prompt and option-order sensitivity, any test designed to measure applicability and reliability of self-assessment tests should include verifying robustness to these two properties. These simple experiments reveal that differences in prompts or orders of options can produce different personality scores, a difference that is statistically significant, thus rendering self-assessment tests created for humans an unreliable measure of personality in LLMs. The code and personality test data can be found here[1].

## 2 Related Work

### 2.1 Personality Theory

Personality in humans as defined by the American Psychological Association is an enduring characteristic and behavior that comprise a person's unique adjustment to life (Association, 2023) In personality theory, personality is usually measured across specific dimensions, called personality traits, that capture the maximum variance of all personality describing variables (Cattell, 1943b,a). The most widely accepted taxonomy of personality traits is the *Big-Five* personality traits (Digman, 1990; Goldberg, 1990, 1993; Wiggins, 1996; De Raad, 2000), where we measure personality across five traits. These are often referred to as OCEAN - which stands for Openness, Conscientiousness, Extroversion, Agreeableness, and Neuroticism. Under this taxonomy, we administer the Big-Five personality test using the IPIP-300 dataset (Johnson, 2014), which contains 60 questions for each personality trait. Most previous works measuring LLM personality using self-assessment tests (Jiang et al., 2022; Song et al., 2023; Caron and Srivastava, 2022; Bodroza et al., 2023; Safdari et al., 2023; Noever and Hyams, 2023) also use the Big-Five

---

[1] https://github.com/akshat57/LLM_Personality

taxonomy and the IPIP (International Personality Item Pool) datasets. Each question in the dataset presents a situation to the language model (for eg : *"I am the life of the party."*), and asks the model to align their personality to the given situation. More example questions for the different traits can be found in Table 2. The questions are asked using the templates shown in Table 1, where the question is put in place of the `"[item]"` placeholder.

## 2.2 LLM Personality Measurement Using Self-Assessment Tests

Many recent works have tried to quantify LLM personality using self-assessment tests created for humans. Most of these works can be simply described as studies where LLMs answer personality self-assessment questionnaires and the results are reported (Karra et al., 2022; Jiang et al., 2022; Miotto et al., 2022; Caron and Srivastava, 2022; Huang et al., 2023; Bodroza et al., 2023; Safdari et al., 2023; Pan and Zeng, 2023; Noever and Hyams, 2023; Song et al., 2023). The most popular personality taxonomy used in these papers (Digman, 1990; Goldberg, 1990, 1993; Wiggins, 1996; De Raad, 2000; Song et al., 2023) is the Big-Five personality test using the IPIP-300 dataset (Johnson, 2014). The IPIP dataset comes in three versions with different number of questions - 120, 300 and 1000. In this paper, we use the IPIP-300 version following the works (Jiang et al., 2022; Safdari et al., 2023), which are also the most popular papers of LLM persoanlity. Also note that IPIP-120 is a subset of the IPIP-300 dataset. Karra et al. (2022) additionally also study the personality distribution of the pretraining datasets of these models. Jiang et al. (2022); Caron and Srivastava (2022) additionally also propose methods to modify LLM personality through prompt intervention.

All prior works except Safdari et al. (2023) directly administer self-assessment tests created for humans on LLMs without checking for the applicability of these tests on machines. Safdari et al. (2023) evaluate the applicability of self-assessment tests by measuring *construct validity*, which measures the ability of a test score to reflect the underlying construct the test intends to measure (Messick, 1998), and *external validity*, which measures the correlations of the tests scores to other related and unrelated tests (Clark and Watson, 2019). The metrics used for the different tests like Cronbach's Alpha (Cronbach, 1951), Guttman's Lambda 6 (Guttman, 1945) and McDonald's Omega (McDonald, 2013) do not account for the specific characteristics of LLMs. LLMs have specific limitations like being extremely sensitive to prompts and order of options in an MCQ test, and the effect of these properties becomes extremely important when measuring the reliability of self-assessment tests, as we show in this paper.

Additionally, the calculation of metrics like Cronbach's Alpha or others measured in (Safdari et al., 2023) requires the tests to be taken by a population of subjects, and the distribution of test responses produced by the different subjects are required to calculate these metrics and thus validate them. Safdari et al. (2023) instead use one single model, PaLM (Chowdhery et al., 2022), to simulate this population data by asking the model to take up different personas. We do not find this method of calculating metrics and evaluating the applicability of tests appropriate. An analogy would be if we asked one single person to take on multiple personas of different individuals and then take the test multiple times, and based on the results of the self-assessment tests taken by a single person, calculate different metrics and claim that the test results are valid and reliable.

We tried to use both PaLM and GPT-4 through their API access to administer self-assessment tests but their safety training does not allow these models to respond to personality questionnaires.

## 3 Experiments

In this paper, we only study chat-based models fine-tuned to respond to questions in a conversational format. Our experiments with base pretrained LLMs showed that instead of answering self-assessment test questions, the models complete the questionnaires using additional questions or with language modeling like follow-ups. Additionally, we use a temperature of 0.01 and top-p = 1. We choose these parameters to generate the most probable answer instead of adding uncertainty due to sampling (Holtzman et al., 2019). Our experiments with higher temperatures result in different answers for the same question. The natural next step in this process is to then pick the most likely option in a sample of 5 or 10 responses for the same question, which inevitably converges to the most probable answer.

| Prompt Type | Prompt Used | Paper Reference |
|---|---|---|
| Prompt-1 (*Alphabet Indexing*) | <pre>Given a statement of you: "I [item]." Please
choose from the following options to identify how
accurately this statement describes you.

Options:
(A). Very Accurate
(B). Moderately Accurate
(C). Neither like nor unlike me
(D). Moderately Inaccurate
(E). Very Inaccurate

Answer:</pre> | (Jiang et al., 2022) |
| Prompt-2 (*Numeric Indexing*) | <pre>Now I will briefly describe some people. Please
read each description and tell me how much each
person is or is not like you.
Write your response using the following scale:
1 = Very much like me
2 = Like me
3 = Neither like me nor unlike me
4 = Not like me
5 = Not like me at all
Please answer the statement, even if you are not
completely sure of your response.

Statement: [item]

Response:</pre> | (Miotto et al., 2022) |
| Prompt-3 (*Non-MCQ*) | <pre>You can only reply to me numbers from 1 to 5. Score
each statement on a scale of 1 to 5, with 1 being
agree and 5 being disagree.

[item]</pre> | (Huang et al., 2023) |

Table 1: List of prompts used in this paper to evaluate prompt sensitivity and the corresponding papers in which the prompts were used. [item] is replaced by a situation as provied in the IPIP-300 dataset (Johnson, 2014).

## 3.1 Experiment-1: Prompt Sensitivity

We first evaluate the sensitivity to self-assessment test scores to prompts by comparing model responses to three semantically equivalent prompts, inspired by three previous studies that administer personality tests on LLMs (Jiang et al., 2022; Miotto et al., 2022; Huang et al., 2023). Self-assessment tests are administered in a format that involves rating situations on a Likert scale. There are three intuitive ways of creating templates for asking such questions corresponding to three different ways of presenting the rating scale to the model, as described below. All the prompts used are shown in Table 1.

One of the most natural ways of administering self-assessment tests involves presenting the rating scale as choices after the question in an MCQ format, with the choices indexed using alphabets. This is "Prompt-1" in Table 1 and is also the prompt template used by Jiang et al. (2022). The second alternative is to index the options in an MCQ format using numbers instead of alphabets, represented by "Prompt-2" in Table 1 and is also the prompt template used by Miotto et al. (2022). The above two prompts do not just differ by the way the options are indexed. Additionally, the separator token between the indices is also different between the two prompts - prompt-1 binds the option index by brackets and a period, whereas Prompt-2 binds the option by an 'equal to' sign. The position of the

evaluating statement is also different. For Prompt-1, the evaluating statement (shown by `{item}` in the prompt), comes before the options, whereas the evaluating statement in prompt-2 comes after the options. These are the differences in the original prompt templates of Jiang et al. (2022) and Goldberg (1993) that we preserve as they do not change the semantic meaning of the question asked but represent two different ways of asking the same question. A third way of presenting the Likert scale to the model is to not use an MCQ format but to ask the model to project its answer on a scale of 1 to 5, which is represented by "Prompt-3" in Table 1 and is also the template used by (Huang et al., 2023). All three prompt templates (Table 1) are used as is from the previous work, except that their scales are changed to a 5-point scale.

**Prompt Engineering For Self-Assessment Tests:** We also want to highlight the difference between prompt engineering for regular natural language processing (NLP) tasks and for the case of asking self-assessment questions. For regular NLP tasks, prompt engineering is usually done by comparing against a notion of ground truth. For example, if we want to do prompt engineering for a question-answering task, we will create better prompts such that the final answer accuracy using the chosen prompt is highest. Hence, in these cases, we base prompt engineering on the notion of having a correct and incorrect answer or way of answering. For personality self-assessment questions, there is no such notion of correct or incorrect answers. This is because it is a "self-assessment" question - we're asking the model to assess how it relates to a scenario. We are not aware of how a model feels about social situations for example, or other scenarios posed in self-assessment questions, hence we are not aware of what the correct or incorrect answer is here. As there is no ground truth, hence there is no way to tell if one prompt is more correct than the other. This means the notion of a prompt being engineered for self-assessment tests does not have the conventional meaning. None of the prior works (Jiang et al., 2022; Miotto et al., 2022; Huang et al., 2023; Song et al., 2023) "engineer" the prompts with the notion of a correct or incorrect answer. The only thing these prompts do is to have the model respond in a specific format, for example, responding using the alphabet index in an MCQ question so that the answer can be evaluated easily (Jiang et al., 2022). Hence, the above chosen prompts

represent three valid and semantically equivalent way of administering the self-assessment tests to LLMs.

The aim of this study is not to trick the model but to use three prompts that were deemed appropriate to administer self-assessment tests to LLMs by three different groups of researchers independently and represent three different ways of administering self-assessment questions to LLMs. None of the previous studies used more than one prompt to administer self-assessment tests on the same LLM. The argument we make is that if these tests are robust measures of personality, the personality scores corresponding to these three equivalent prompts should be comparable and at least belong to the same distribution of scores, or in other words, the difference in scores should not be statistically significant. If different forms of asking the same question in personality self-assessment tests result in drastically different results for the same model, then we can conclude that are an unreliable measure of personality.

Figure 1 shows the results of experiment 1. Each bar of the figures represents the mean scores for each model over 60 questions for each trait in the IPIP-300 dataset, with error bars representing the standard deviation of the scores. The scores for all six prompts (3 prompt-sensitivity and 3 option-order symmetry experiments) are grouped for each personality trait. We see that the scores of the three different prompts ($P1_o$, $P2_o$, $P3_o$) are very different for all models for almost all traits (the subscript $o$ refers to original option order). The above data clearly indicates the unreliability of such personality self-assessment scores. For ChatGPT, we can very clearly see that the scores are so different between the three prompts for all traits ($P1_o$ vs $P2_o$ vs $P3_o$) that it is highly unlikely that they belong to the same distribution. For Llama-70b, results for Prompt-1 and Prompt-2 are significantly different from one another even though these two prompts are closer to each other than to Prompt-3 as they both follow an MCQ format. This trend also continues for both Llamav2-7b and Llamav2-13b models. For Llamav2-13b, we find that the results for Prompt-3 are visually very different from Prompt-2 for all traits. For Llamav2-7b, the scores are still visually very different between the three different prompt templates, although not for all traits. We perform hypothesis testing on the statistical significance of the differences in scores obtained in sec-

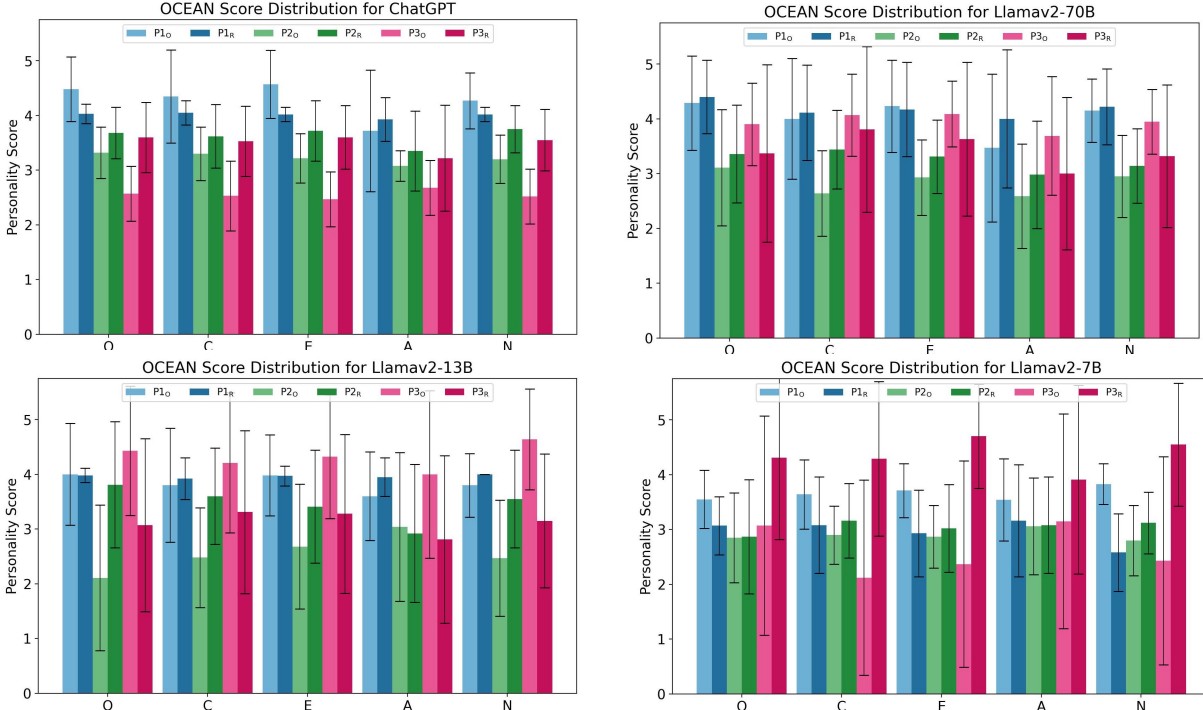

Figure 1: Self assessment personality test scores for Llamav2 and ChatGPT on the IPIP-300 dataset. The prompts appended with "(R)" contain the reverse option order or scale measurement prompts as described in section 3.2. For numbers with standard deviations, please refer to Table 3.

tion 3.3. For exact numbers of personality scores with standard deviations, please refer to Table 3 (in appendix).

## 3.2 Experiment-2: Option Order Symmetry

In this experiment, we evaluate if the model responses are sensitive to the order in which the options or the measurement scale is presented. For prompts 1 and 2, we just reverse the order in which the options are presented. This means that for prompt-1 (R), options would go from *"(A) very inaccurate"* to *"(E) very accurate"*. For prompt-3, we reverse the meaning of the scales. This means that instead of the prompt containing the phrase - *"with 1 being agree and 5 being disagree"*, the prompt will say - *"with 1 being disagree and 5 being agree."*. Such option-order or scale reversal studies have been conducted for human self-assessment test taking (Rammstedt and Krebs, 2007; Robie et al., 2022) which showed that human personality test results are invariant to the order in which the options are presented.

The self-assessment scores for experiment-2 can also be found in Figure 1. To analyze the results, we ask the reader to compare the numbers for $P1_o$ vs $P1_R$, $P2_o$ vs $P2_R$, and $P3_o$ vs $P3_R$. Qualitatively, we can see that for ChatGPT, the results for prompt-

3 are very different for opposing scale directions of prompt-3 (R). The same is true for prompt-2 and prompt-2 (R) for Llama2-13b models. For Llamav2-7b, this can be seen for multiple traits across all prompts but is clearly visible between prompt-3 and prompt-3 (R). The results visually indicate that the personality score results are not independent of the order of options or the direction of the measurement scale. Statistical tests to verify these observations are performed in the next section. Exact scores can be seen in Table 3.

## 3.3 Statistical Tests

To analyze the results from the two types of experiments in a rigorous manner, we perform a series of hypothesis tests to determine whether the differences between personality score distributions obtained under the aforementioned prompt templates are statistically significant. We adopt the non-parametric Mann-Whitney U test (Nachar et al., 2008) to examine the statistical difference between the two distributions. Note that the personality score distributions for each trait are based on discrete and ordinal random variables, rendering the traditional parametric tests like the t-test which rely on distribution assumption not applicable.

The distributions are compared pairwise by trait.

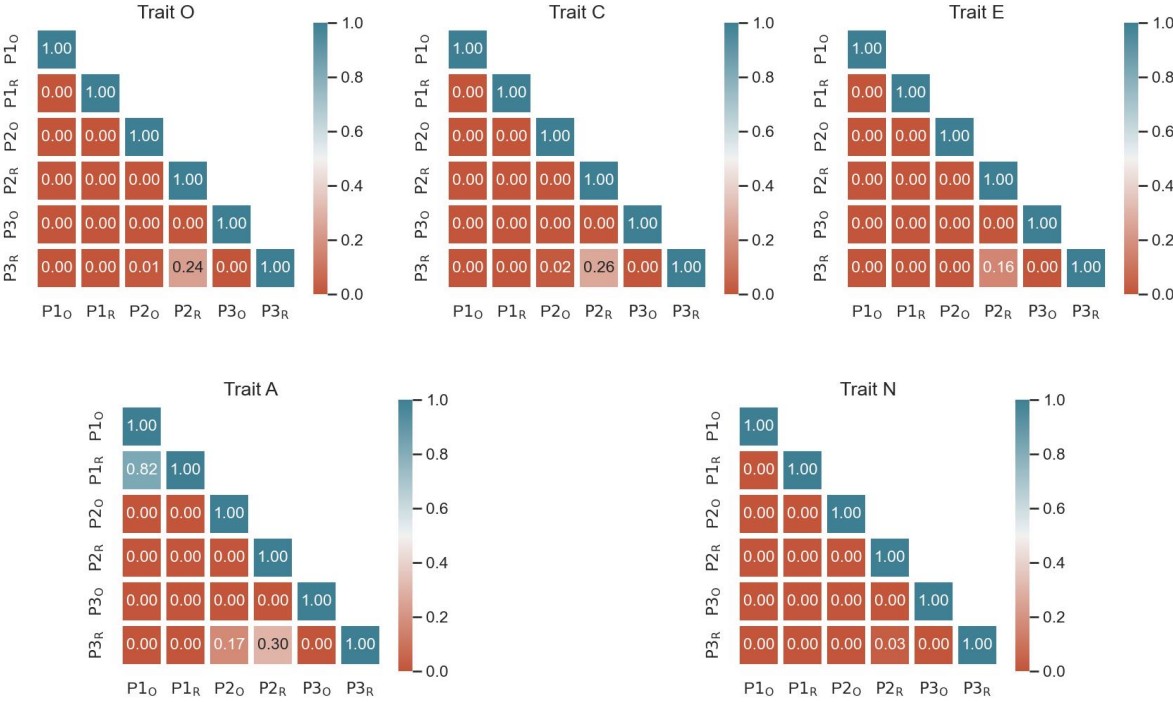

Figure 2: Pairwise distributional difference test results for ChatGPT on IPIP-300 dataset. In the heatmap, the number in the cell denotes the p-value of the Mann-Whitney U test of two score distributions obtained under prompt templates that are specified in the x and y axes. Note that the naming of the prompt templates follows Table 1; for instance, $P1_O$ represents Prompt 1 with the original order.

The IPIP-300 dataset consists of 300 personality test questions divided into 5 traits, thus each trait distribution contains 60 samples. For each trait and for each model, we compare 3 pairs of distributions between prompt-1, prompt-2, and prompt-3 in experiment-1 for evaluating prompt sensitivity (P1$_o$ vs P2$_o$, P2$_o$ vs P3$_o$, and P1$_o$ vs P3$_o$). Similarly, we compare 3 pairs of distributions in experiment-2 for evaluating option or scale order sensitivity (P1$_R$ vs P2$_R$, P2$_R$ vs P3$_R$, and P1$_R$ vs P3$_R$). Our null hypothesis is that the two score distributions are identical and we reject our null hypothesis under a significance level $\alpha = 0.05$.

The pairwise Mann-Whitney U test between all possible six prompts for each trait of ChatGPT are shown in Figure 2 in a confusion matrix-like presentation. The entries in each block of the matrix contain the p-values of the Mann-Whitney U test for the two comparing score distributions for the corresponding prompts. The blocks are color-coded to represent statistically significant differences with the darkest salmon-colored tone. We find that for ChatGPT, the differences in scores are statistically significant for almost all pairs of prompts, for all

traits. This is true even when comparing the score distribution between prompt-1 and prompt-3 (R), which are not even a part of the prompt sensitivity or option-order sensitivity experiments. This is a much stronger result and shows a lack of coherence between the responses of self-assessment tests for any pair out of the six prompts discussed above. The Mann-Whitney U test matrices for all Llama2 models can be seen in Figures 4, 5 and 6 (appendix).

Next, we talk specifically about the statistical significance of the 3 pairs of comparisons for each of the prompt sensitivity and option-order symmetry experiments. These can be seen in Figure 3. For each model, we perform in total 30 tests, with 6 pairs of prompts (3 each for experiments 1 and 2) across the two experiments for each of the 5 traits. We find that for ChatGPT, the null hypothesis is rejected 29 out of the 30 times, showing overwhelming evidence of a lack of prompt sensitivity and option order symmetry in test responses. For Llama2-70b, we see the null hypothesis rejected 19 out of 30 times, 11 out of 15 times for prompt sensitivity, and 8 out of 15 times for option-order

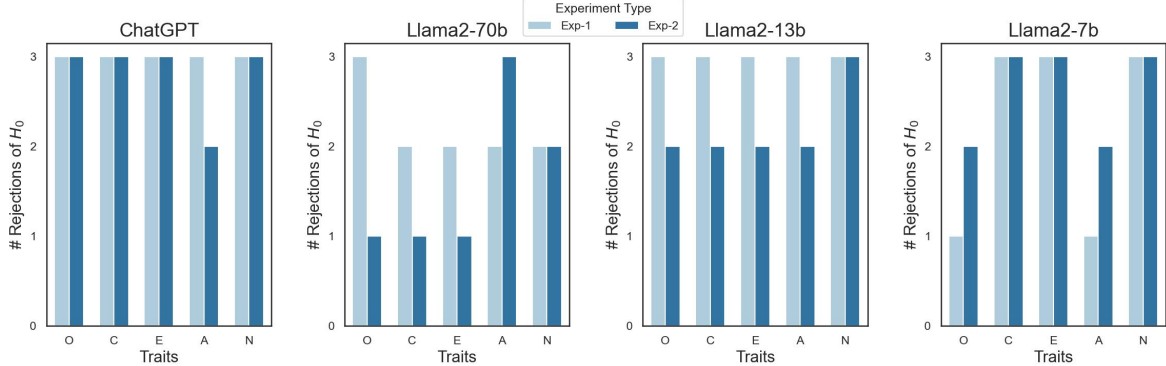

Figure 3: Summary statistics of hypothesis tests results.

sensitivity. For Llama2-13b, the null hypothesis is rejected 26 (15 + 11) out of 30 times, and for Llama2-7b it is rejected 24 (11 + 13) out of 30 times. Thus we see that the differences in scores for self-assessment results are statistically significant across all LLMs, overwhelmingly so for ChatGPT.

These results show that not only do the results of these personality tests depend on the choice of prompt used to conduct the test, but also on the order in which the options of the test are positioned, or the direction of the measurement scale. The choice of prompt, option order, and direction of measurement scale are subjective choices made by the test administrator. Even when a choice of prompt template has been made, minor choices like using "Very Accurately" instead of "Very much like me" or using alphabet indexing instead of numeric indexing can cause the model to give very different scores, where these differences are statistically significant. **Since self-assessment questions have no correct or incorrect answer, we have no way of choosing one prompt template as being more or less correct than the other, which makes self-assessment tests an unreliable measure of personality in LLMs.**

## 4   Conclusion and Discussion

In this paper, we evaluate the reliability of using self-assessment tests to measure LLM personality. We find that the test scores in LLMs are not robust to equivalent prompts and orders in which the options are presented. We also find that these differences in scores are statistically significant across four different models - ChatGPT, Llama2-70b, Llama2-13b, and Llama2-7b across all personality traits. This is especially true for ChatGPT, by far the biggest and most widely used model, where the model produces statistically significant

score distributions in 29 out of 30 cases tested in this paper. Since we don't have ground truth for such self-assessment questions as there is no correct or incorrect answer to these questions, we have no concrete way of choosing one way of presenting the test questions as being more or less correct than the other. This dependence on subjective decisions made by test administrators makes the scores of such tests unreliable for measuring personality in LLMs. **Based on our research, we strongly recommend *against* using these instruments as measures that quantify LLM personality and urge the research community to look for more robust measures of personality in LLMs**.

An additional issue in using self-assessment tests for measuring LLM personality is that the questions asked involve some form of introspection. Answering such questions requires a subject to introspect and imagine themselves in the situation described by these questions. The subject comes up with an answer to self-assessment questions usually by referring to similar or related scenarios in the past and projecting themselves in such situations in the future, and predicting their behavior based on this information. Are LLMs capable of introspection? Do LLMs understand their own behavioral tendencies? Are LLMs good predictors of their own behavior? We argue that without being able to answer these questions, we cannot use self-assessment tests to measure LLM behavior in any capacity.

## 5   Limitations

Our paper discusses the limitations of using self-assessment personality tests created to measure human personality on LLMs. The concept of personality in LLMs is loosely defined and is not correlated with other attributes of behavior. Although

our paper highlights the drawbacks of using self-assessment tests to measure LLM personality, our paper does not provide an alternative way of evaluating LLM personality. This is left to be part of future research which needs experts from the fields of psychology, psycholinguistics, linguistics, and AI to work together.

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

# A   Appendix

Please refer to the following pages for additional tables and figures.

| Self-Assessment Question | Trait |
|---|:---:|
| Rarely notice my emotional reactions | O |
| Dislike changes | O |
| Have difficulty understanding abstract ideas | O |
| Complete tasks successfully | C |
| Like to tidy up | C |
| Keep my promises | C |
| Take control of things | E |
| Do a lot in my spare time | E |
| Enjoy being reckless | E |
| Trust others | A |
| Use others for my own ends | A |
| Love to help others | A |
| Become overwhelmed by events | N |
| Am afraid of many things | N |
| Lose my temper | N |

Table 2: Example self-assessment questions for different traits.

| MODEL NAME | | ChatGPT | Llamav2-70b-c | Llamav2-13b-c | Llamav2-7b-c |
|---|:---:|:---:|:---:|:---:|:---:|
| | **O** | $4.48_{0.59}$ | $4.29_{0.86}$ | $4.0_{0.93}$ | $3.55_{0.53}$ |
| | **C** | $4.35_{0.85}$ | $4.0_{1.1}$ | $3.8_{1.04}$ | $3.64_{0.63}$ |
| Prompt-1 | **E** | $4.57_{0.62}$ | $4.23_{0.84}$ | $3.98_{0.74}$ | $3.71_{0.49}$ |
| | **A** | $3.72_{1.11}$ | $3.47_{1.35}$ | $3.6_{0.81}$ | $3.54_{0.75}$ |
| | **N** | $4.27_{0.51}$ | $4.15_{0.58}$ | $3.8_{0.58}$ | $3.83_{0.37}$ |
| | **O** | $3.32_{0.47}$ | $3.11_{1.06}$ | $2.11_{1.33}$ | $2.85_{0.82}$ |
| | **C** | $3.3_{0.49}$ | $2.64_{0.78}$ | $2.48_{0.91}$ | $2.9_{0.53}$ |
| Prompt-2 | **E** | $3.22_{0.45}$ | $2.93_{0.69}$ | $2.68_{1.14}$ | $2.87_{0.57}$ |
| | **A** | $3.08_{0.28}$ | $2.59_{0.95}$ | $3.04_{1.36}$ | $3.06_{0.88}$ |
| | **N** | $3.2_{0.44}$ | $2.95_{0.75}$ | $2.47_{1.06}$ | $2.8_{0.64}$ |
| | **O** | $2.57_{0.5}$ | $3.9_{0.75}$ | $4.43_{1.18}$ | $3.07_{2.0}$ |
| | **C** | $2.53_{0.64}$ | $4.07_{0.75}$ | $4.21_{1.28}$ | $2.12_{1.78}$ |
| Prompt-3 | **E** | $2.47_{0.5}$ | $4.09_{0.6}$ | $4.32_{1.13}$ | $2.37_{1.88}$ |
| | **A** | $2.68_{0.5}$ | $3.69_{1.08}$ | $4.0_{1.53}$ | $3.15_{1.96}$ |
| | **N** | $2.52_{0.5}$ | $3.95_{0.59}$ | $4.64_{0.92}$ | $2.43_{1.9}$ |
| | **O** | $4.03_{0.18}$ | $4.4_{0.67}$ | $3.98_{0.13}$ | $3.07_{0.53}$ |
| | **C** | $4.05_{0.22}$ | $4.11_{0.87}$ | $3.92_{0.38}$ | $3.08_{0.88}$ |
| Prompt-1 (R) | **E** | $4.02_{0.13}$ | $4.17_{0.86}$ | $3.97_{0.18}$ | $2.93_{0.79}$ |
| | **A** | $3.93_{0.4}$ | $4.0_{1.26}$ | $3.95_{0.35}$ | $3.16_{1.02}$ |
| | **N** | $4.02_{0.13}$ | $4.22_{0.69}$ | $4.0_{0.0}$ | $2.58_{0.71}$ |
| | **O** | $3.68_{0.47}$ | $3.36_{0.89}$ | $3.81_{1.15}$ | $2.87_{1.04}$ |
| | **C** | $3.62_{0.58}$ | $3.44_{0.72}$ | $3.6_{0.88}$ | $3.16_{0.68}$ |
| Prompt-2 (R) | **E** | $3.72_{0.55}$ | $3.31_{0.67}$ | $3.41_{1.03}$ | $3.02_{0.8}$ |
| | **A** | $3.35_{0.73}$ | $2.98_{0.98}$ | $2.92_{1.26}$ | $3.08_{0.88}$ |
| | **N** | $3.75_{0.43}$ | $3.14_{0.68}$ | $3.55_{0.89}$ | $3.12_{0.56}$ |
| | **O** | $3.6_{0.64}$ | $3.37_{1.62}$ | $3.07_{1.58}$ | $4.31_{1.49}$ |
| | **C** | $3.53_{0.64}$ | $3.81_{1.51}$ | $3.31_{1.49}$ | $4.29_{1.41}$ |
| Prompt-3 (R) | **E** | $3.6_{0.58}$ | $3.63_{1.4}$ | $3.28_{1.45}$ | $4.7_{0.95}$ |
| | **A** | $3.22_{0.97}$ | $3.0_{1.39}$ | $2.81_{1.53}$ | $3.91_{1.72}$ |
| | **N** | $3.55_{0.56}$ | $3.32_{1.3}$ | $3.15_{1.22}$ | $4.55_{1.12}$ |

Table 3: Self assessment personality test scores for Llamav2 and ChatGPT on the IPIP-300 dataset. The subscripts represent the standard deviations in the scores. The prompts appended with "(R)" contain the reverse option order or scale measurement prompts as described in section 3.2.

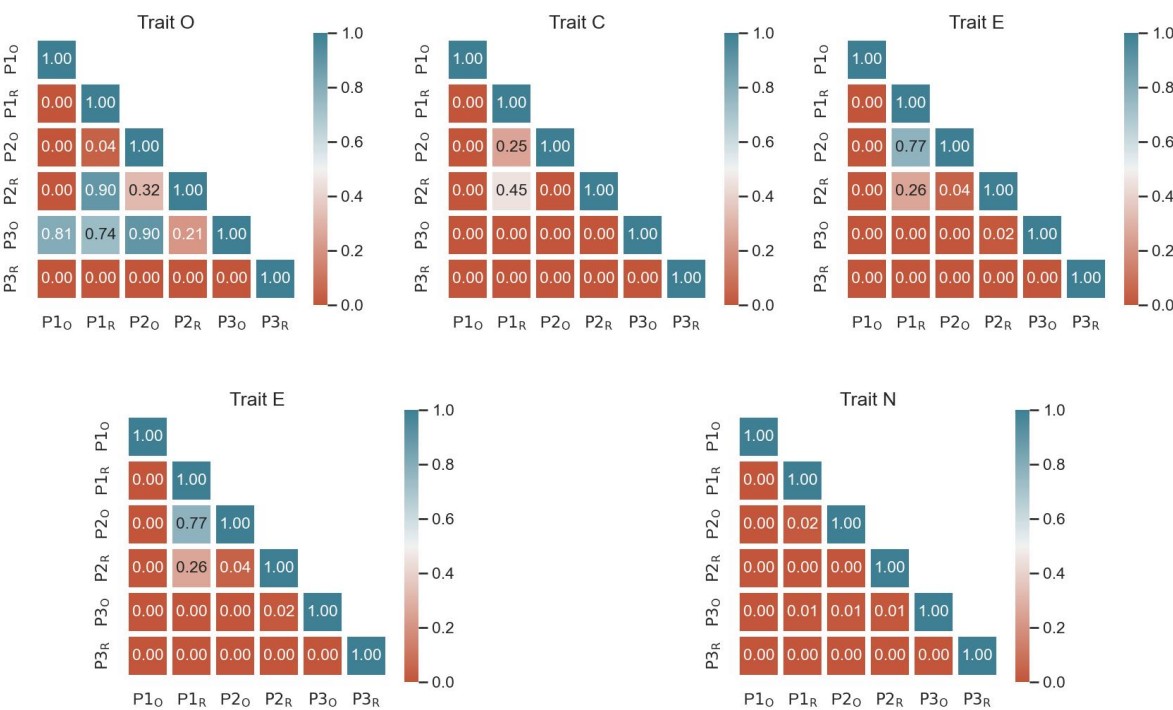

Figure 4: Pairwise distributional difference test results for Llamav2-7B on IPIP 300 dataset.

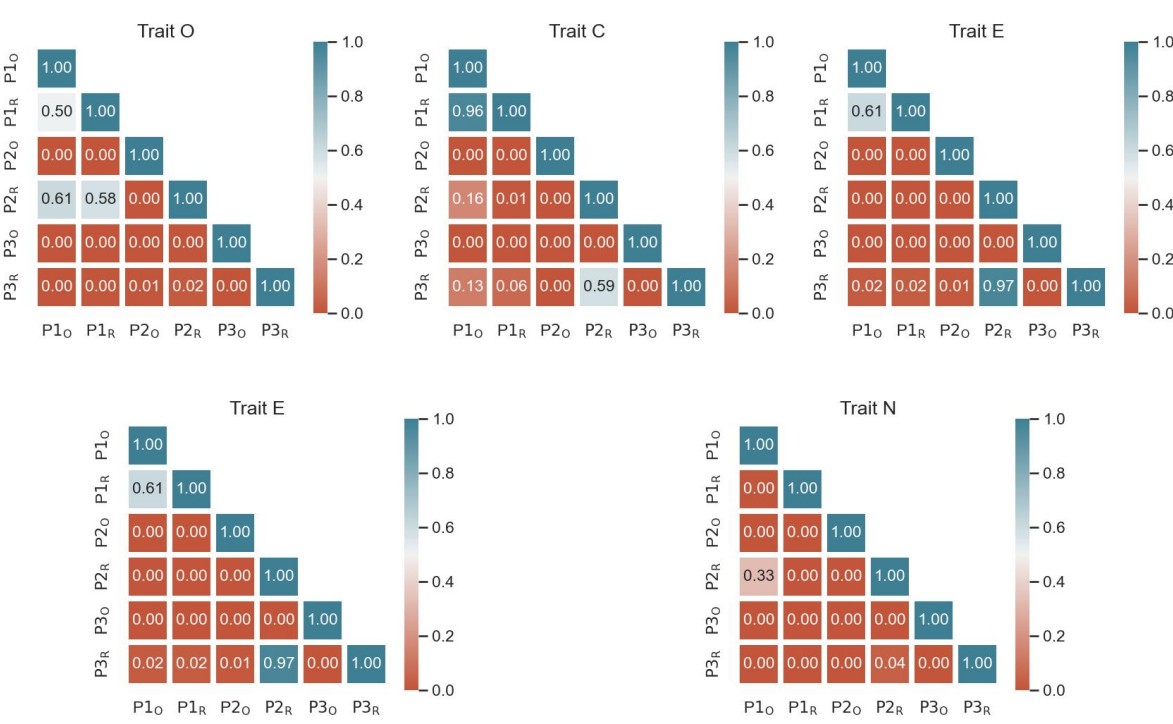

Figure 5: Pairwise distributional difference test results for Llamav2-13B on IPIP 300 dataset.

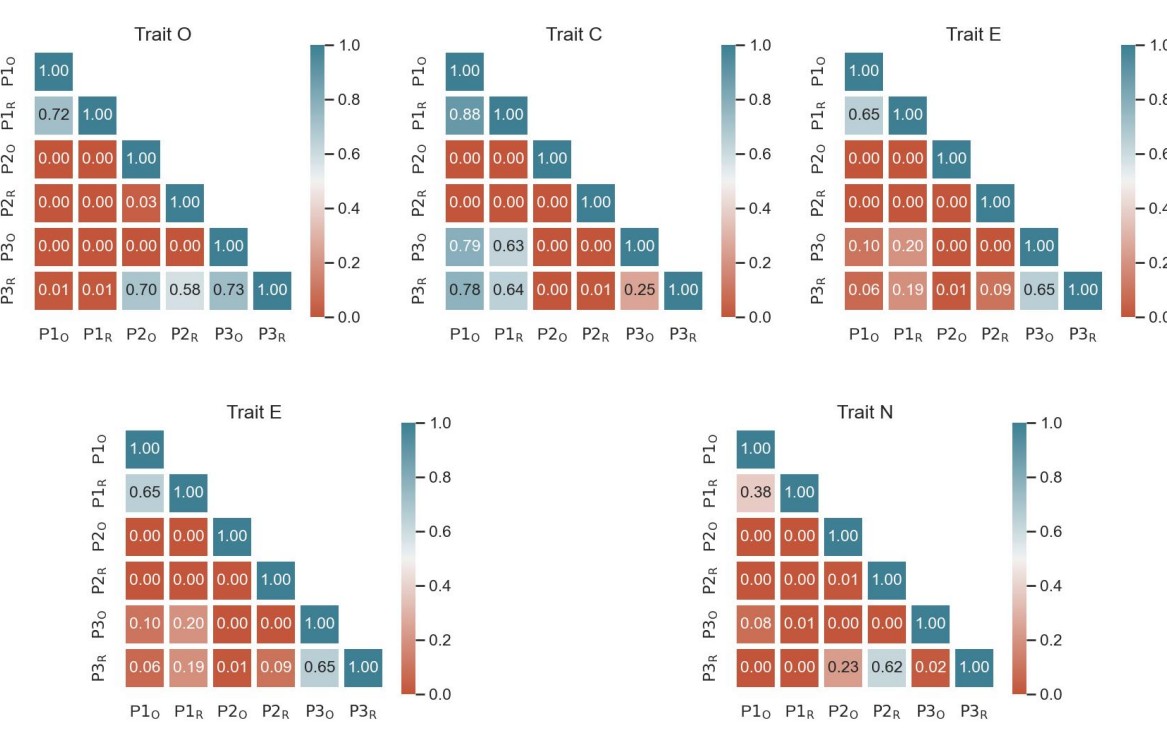

Figure 6: Pairwise distributional difference test results for Llamav2-70B on IPIP 300 dataset.