# OpenReview forum: "Self-Assessment Tests are Unreliable Measures of LLM Personality"
_EMNLP/2024/Workshop/BlackBoxNLP — BlackboxNLP 2024_

### Official Review · Reviewer_MU7f · 2024-09-02

**Overall Assessment:** 4
**Confidence:** 3

**Best Paper:**

1

**Best Paper Justification:**

-

**Comments Questions Suggestions And Typos:**

I could not understand the end of section 2.2 "Additionally, the calculation... " consider rephrasing

You might want to refer also to:

Mizrahi, M., Kaplan, G., Malkin, D., Dror, R., Shahaf, D., & Stanovsky, G. (2024). State of what art? a call for multi-prompt llm evaluation. Transactions of the Association for Computational Linguistics, 12, 933-949.

Natalie Shapira, Mosh Levy, Seyed Hossein Alavi, Xuhui Zhou, Yejin Choi, Yoav Goldberg, Maarten Sap, and Vered Shwartz. 2024. Clever Hans or Neural Theory of Mind? Stress Testing Social Reasoning in Large Language Models. In Proceedings of the 18th Conference of the European Chapter of the Association for Computational Linguistics (Volume 1: Long Papers), pages 2257–2273, St. Julian’s, Malta. Association for Computational Linguistics.

**Paper Summary:**

This paper, "Self-Assessment Tests are Unreliable Measures of LLM Personality," provides a critical review of the ability to measure models' personalities.
The authors used the Big-Five personality test using the IPIP-300 dataset and showed that models are inconsistent in their answers when rephrasing the instruction in the prompt or the order of the answers (as opposed to humans who, according to the literature, are consistent).

**Summary Of Strengths:**

The paper is clear and seems to be well-based.

The motivation for conducting the experiments is clear.

The results of the experiments are important to raise awareness in the community.

The discussion raises important points.

**Summary Of Weaknesses:**

The scope of the study is relatively small. Given the fact that it was sent to a workshop, I think its OK.

There is a lack of consideration for more complex situations, such as how a language model behaves when its personality is fixed in the prompt. Perhaps the reason for the varying results is that it's currently not initialized to anything and therefore, in a certain sense, random.

It's important to evaluate not just the consistency of a model's specific answers, but also the overall consistency of its behavior. We should be cautious about applying human-centric testing methods to AI models, as their underlying mechanisms and behaviors can differ significantly from human cognition. Therefore, we need to reassess the validity of traditional testing approaches when applied to these artificial systems (Shapira et al., 2023).

---

### Official Review · Reviewer_Baox · 2024-09-09

**Overall Assessment:** 4
**Confidence:** 3

**Best Paper:**

1

**Best Paper Justification:**

No

**Comments Questions Suggestions And Typos:**

- Maybe a more promising path for research are some more intricate test than direct self-assessment? I am no expert in psychology, but I assume self-assessment is only good if the participant is willing to take part and to truthfully answer the test questions. I assume that there are probably other kinds of tests for non-collaborating individuals, such as people who want to deceive the examiner, or people who are lying, or people just answering randomly, or people with a poor capability of truthfully assessing themselves and thus reporting their subjective image of themselves instead of their real selves. I would believe there are psychological tests and methods applicable in such settings and could be a great inspiration for the problem at hand, as this seems closer to how LLMs operate and "behave". I would imagine some indirect tests, which ask you to act in some way without you knowing what is being tested so that you uncounsciously exhibit some aspects of your personality.
- Some questions of the test seem to be really meaningless for a LLM which is not embodied and has no agency in real world (such as how fun they are at parties, or how tidy they are...) I understand the paper just takes over the approach from previous works to disprove it, but I still find that this is an important aspect of the test, that it assumes that the individual actually does actively interact with the world.
- The presentation in section 3 should be reordered a bit. First, the experiments should be explained, and only then evaluated. (E.g. now line 384 and further already partially discusses the results of Experiment 2, which is only explained later on line 416.)

**Paper Summary:**

The paper examines whether self-assessment personality tests are any good for determining a "LLM personality" by having the LLM to answer the test questions, as has been done by numerous previous studies. The paper reliably shows that LLMs are sensitive to random unimportant changes in the presentation of the questions, leading to significantly different results of the test, thus proving that this method is unreliable.

**Summary Of Strengths:**

- Simple and systematic method
- Confident disproval of a range of previous papers
- Explicitly reviews how studies have shown the validity of personality tests on humans but similar studies had not been performed for LLMs.

**Summary Of Weaknesses:**

- I am completely missing a vital discussion of what an assumed "LLM personality" is and to what extent it even makes sense to assume a personality of a LLM, let alone measure it. The paper indirectly has a reserved stance towards this, simply disproving previous papers which had assumed a LLM personality, but not really delving deep into this concept by itself. I would love to have at least a clear position of the authors towards this, i.e. whether they assume it makes sense to talk about a "LLM personality", if yes then based on what and how it would be defined and how it potentially could be tested; or if not, then why.
- The paper examines only one self-assessment test in a few settings but concludes (in the abstract) that they "show that self-assessment personality tests created for humans are unreliable measures of personality in LLMs". This is definitely an overgeneralization, you have not shown this in general.
- The discussion about missing ground truth and thus inability to test whether the test returns correct results is missing comparison to the situation in humans. How did we do that in humans so that we check whether the personality tests reveal the true personality? (This is not only about a missing reference, this is about looking for a more principled way to validate the testing method.)

---

### Official Review · Reviewer_py3M · 2024-09-09

**Overall Assessment:** 3
**Confidence:** 4

**Best Paper:**

1

**Best Paper Justification:**

NA

**Comments Questions Suggestions And Typos:**

I am personally not a fan of using human-inspired tests to tests generative models. What do we learn from this type of tests/experiments that is useful and actionable?

**Paper Summary:**

This paper studies the lack of robustness of LLMs for personality tests with respect to prompt reformulation and reordering of multiple-choice options in the answer. These issues have been studied in more general context and this work makes it specific to the personality tests context. The results of the analysis show large differences in answers when prompts and/or choice orders are varied.

**Summary Of Strengths:**

Confirming the lack of robustness of models wrt prompt paraphrasing and option ordering in the contexts of personality tests

**Summary Of Weaknesses:**

This phenomenon was studied before in different contexts. As such, I don't find the paper novel or insightful.

---

### Decision · Program_Chairs · 2024-09-20

**Decision:**

Accept

**Comment:**

The reviewers overall provided a positive rating for this paper, and I think the paper makes a concrete contribution in that it serves to cast doubt on the credibility of existing work. However, I do empathize with the concerns raised by reviewers py3M and Baox in that, there is a general issue of whether it even makes sense or if it is useful in any sense to subject LLMs to "personality tests" in the first place. The paper stands to point out methodological flaws in other work in the literature that does this, but it may be useful to take an actual stance on this more fundamental issue. Furthermore, I agree with the reviewers that the problem being pointed out itself (effect of MCQ ordering) is a point that has been made many times in the literature, and showing it again in a single task setting here doesn't seem to be a major contribution; the main contribution, in my opinion, is that the paper serves to weaken claims in existing work that may contribute to, e.g., excessive anthromorphization of LLMs.